# Characteristics of Physical Exercise Programs for Older Adults in Latin America: A Systematic Review of Randomized Controlled Trials

**DOI:** 10.3390/ijerph18062812

**Published:** 2021-03-10

**Authors:** Eduardo Vásquez-Araneda, Rodrigo Ignacio Solís-Vivanco, Sandra Mahecha-Matsudo, Rafael Zapata-Lamana, Igor Cigarroa

**Affiliations:** 1Programa de Magister en Fisiología Clínica del Ejercicio, Universidad Mayor, Santiago 8580000, Chile; vasquez.eduardo@hotmail.cl (E.V.-A.); rsolvivanko.10@gmail.com (R.I.S.-V.); 2Facultad de Ciencias, Universidad Mayor, Santiago 8580000, Chile; sandra.mahecha@umayor.cl; 3Escuela de Educación, Universidad de Concepción, Los Ángeles 4440000, Chile; rafaelzapata@udec.cl; 4Escuela de Kinesiología, Facultad de Salud, Universidad Santo Tomás, Los Ángeles 4440000, Chile

**Keywords:** aging, physical aptitude, mental health, cognition, systematic review

## Abstract

Aim: To characterize physical exercise programs for older adults in Latin America. Methods: This review was conducted in accordance with the PRISMA statement. A search for randomized controlled trials (RCTs) published between the years 2015 and 2020 was performed in the Scopus, MedLine and SciELO databases. Results: A total of 101 RCTs were included. A large percentage of the studies had an unclear risk of bias in the items: selection, performance, detection and attribution. Furthermore, a heterogeneous level of compliance was observed in the CERT items. A total sample of 5013 older adults (79% women) was included. 97% of the studies included older adults between 60–70 years, presenting an adherence to the interventions of 86%. The studies were mainly carried out in older adults with cardiometabolic diseases. Only 44% of the studies detailed information regarding the place of intervention; of these studies, 61% developed their interventions in university facilities. The interventions were mainly based on therapeutic physical exercise (89% of the articles), with a duration of 2–6 months (95% of the articles) and a frequency of 2–3 times a week (95% of the articles) with sessions of 30–60 min (94% of the articles) led by sports science professionals (51% of the articles). The components of physical fitness that were exercised the most were muscular strength (77% of the articles) and cardiorespiratory fitness (47% of the articles). Furthermore, only 48% of the studies included a warm-up stage and 34% of the studies included a cool-down stage. Conclusions: This systematic review characterized the physical exercise programs in older adults in Latin America, as well the most frequently used outcome measures and instruments, by summarizing available evidence derived from RCTs. The results will be useful for prescribing future physical exercise programs in older adults.

## 1. Introduction

The aging process of the population is advancing at an accelerated rate and is related to a longer life expectancy [1]. Thus, it is expected that the fraction of the world population over 65 years of age will increase from 9% to 16% by the year 2050 [2]. Latin America is in a similar situation, with a 156% increase in the population of older adults (OAs) [3].

Aging is characterized by physiological changes that, conditioned by extrinsic and intrinsic factors, translate into loss of health, conditioning a decline in the physical and mental skills of OAs [4]. Although current evidence supports the bio-psycho-social benefits of physical activity (PA) and physical exercise (PE), it is known that their practice decreases with age [5]. It is therefore a strong predictor of physical disability [6], associated with an increased risk of mortality. Along these lines, the World Health Organization (WHO) has reported that around 2.3 million deaths each year are due to physical inactivity [7]. The highest levels of physical inactivity in men and women (39%) are registered in Latin America and the Caribbean [8,9,10].

The practice of PA and PE through supervised programs contributes to improving physical fitness components such as cardiorespiratory fitness, muscular strength, gait and balance, and to avoid the risk of falls [11,12,13], being an effective intervention to delay the onset characteristic disorders of OA, such as sarcopenia and/or frailty syndrome, which cause a significant deterioration in functionality and quality of life [14,15,16]. On the other hand, PA and PE also generate positive effects associated with psychosocial and cognitive aspects in OAs, reducing symptoms of anxiety and depression [17].

Current PA and PE recommendations for aging suggest accumulating a minimum of 150 min of moderate aerobic PA or 75 min of vigorous aerobic PA and varied multicomponent physical activities three or more days a week, to improve functional capacity and prevent falls, in addition to perform activities that strengthen the main muscle groups two or more days a week [18,19,20].

At the Latin American level, different countries have proposed guidelines for PA recommendations (GPAR) for OAs [21,22,23,24,25,26,27,28,29,30,31,32,33,34,35,36]. Although the recommendations proposed by the WHO for the elaboration of GPAR have been considered as a reference, these are constantly being updated and differ in specific characteristics such as the type of exercise, frequency and duration, as well as the suggested age for their implementation. In relation to the age group to which these GPAR are directed, there are countries that classify those over 65 as elderly, while in Chile this category begins at 60 years. This allows us to infer that there could be a heterogeneity in the GPAR of the different Latin American countries. Along the same lines, it is of great interest to know if the current evidence that exists regarding the prescription of PA and PE in OAs has the same heterogeneity in characteristics such as the type of exercise, analysis variables, measurement instruments, effects on health outcomes, and risk of bias. In addition, knowing in depth the latest and updated research that is being done in the field of PA and PE and thus having a Latin American map of the programs developed in the last five years can serve as a basis for future guidelines, guides, recommendations or programs that wish to be guided by current evidence.

For this reason, the present systematic review aims to characterize the PE programs for OA in Latin America, focusing on the main characteristics of the interventions developed, participants, types of exercise, effects, variables and instruments used, risk of bias and level of compliance with the Consensus on Exercise Reporting Template (CERT) of the articles included. This will allow to start the discussion on the current state of research on PE for OA in the different Latin American countries.

## 2. Materials and Methods

The systematic review was carried out in accordance with the standards established by the PRISMA statement [36]. The PRISMA checklist can be found in the Appendix A. A systematic review protocol had previously been registered in the PROSPERO repository with the code: CRD42020208833.

### 2.1. Search Strategy for the Identification of Studies

The following databases were reviewed: MEDLINE by PubMed, SCOPUS by ELSEVIER and SciELO. The objective was to identify studies that developed PA and PE interventions for OAs in Latin America. The search covered the period between 2015 and 2020. For the development of the search, the MeSH terms used were: “Exercise”, “Exercise Therapy” and “Aged”, present in the MeSH Database. The search strategy followed the Peer Review of Electronic Search Strategies (PRESS) guidelines [37].

The general search syntax was: (“Exercise” OR “Exercise Therapy”) AND (“Aged”) and it was adapted to each database applying the following filters:(a)PubMed: Type of article: randomized controlled trial, Date of publication: 5 years, Language: English, Spanish and Portuguese, Age: aged (65+ years) and 80 and over (80+ years).(b)Scopus: Exclusion: Medline, Year of publication: 2015 to 2020, Status of publication: final, Type of document: article, Country: Latin American countries, Language: English, Spanish and Portuguese, Keyword: words that are related to the subject under study.(c)SciELO: Country: Brazil, Colombia and Chile, Year of publication: 2015 to 2020, Type of literature: article.

Search strings for all databases are presented in the Appendix A.

### 2.2. Selection of Studies and Inclusion Criteria

All those studies that met the search phrase were considered, and only those that met the following inclusion criteria were selected: (a) Country: interventions developed in countries belonging to Latin America; (b) Sample: people over 60 years of age; (c) Language: English, Spanish and Portuguese; (d) Methodological Design: Randomized controlled clinical trial. No reviews, editorial documents, protocols, or thesis were included. The articles selected by title and abstract had to meet the conditions indicated in Table 1.

### 2.3. Data Extraction

Duplicate articles were removed from the databases using Mendeley. Articles that met the inclusion criteria were selected, and when decisions could not be made considering only the title and abstract of the article, the full text was retrieved (Figure 1). A standardized questionnaire was used and applied by the authors to extract the data from the included articles, to synthesize the evidence. The information extracted included: (a) general characteristics of the studies and of the participants (author, year, initial and final sample, adherence, reasons for withdrawal, age range, sex, health condition, recruitment and place of intervention, (b) main characteristics of the interventions based on physical activity and exercise (duration of the intervention, number of sessions per week, time of the session, responsible professional, type of intervention and components of physical fitness addressed and the time allocated to each one of them); (c) main variables evaluated (physical health, mental health and cognitive abilities); (d) main assessment instruments used (physical health, mental health and cognitive abilities).

### 2.4. Risk of Bias Assessment Tool and Consensus on Exercise Reporting Template (CERT) Assessment Form

The Cochrane “Cochrane Manual of Systematic Reviews of Interventions” tool [38] was used to assess the methodological validity of the studies included in this review, evaluating the risk of bias in each of the items proposed by this manual, detailed as follows: (a) Selection bias, (b) Performance bias, (c) Detection bias, (d) Attrition bias and (e) Reporting bias. Only the item: “Other biases” was not considered. The results of this analysis are presented in Figure 2. In addition, the Consensus on Exercise Reporting Template (CERT) assessment form was added to know the proportion of articles that met the CERT items [39] (Appendix A).

### 2.5. Strategy for Data Synthesis

A synthesis of the main findings of the included studies is provided, related to the interventions developed based on PE for OAs in Latin America. The main information is presented in summary tables. In addition, in the discussion, the most relevant methodological and applicability aspects are analyzed and some suggestions are given for future research, in order to standardize the application of this methodology.

## 3. Results

### 3.1. Literature Search

Figure 1 shows the flow chart of systematic reviews proposed by the PRISMA statement. 4.642 potential studies on PA and exercise in OAs in Latin America were identified. After the exclusion of the duplicates in the databases, the screening and eligibility criteria were applied. Finally, for data synthesis, 101 articles were included [40,41,42,43,44,45,46,47,48,49,50,51,52,53,54,55,56,57,58,59,60,61,62,63,64,65,66,67,68,69,70,71,72,73,74,75,76,77,78,79,80,81,82,83,84,85,86,87,88,89,90,91,92,93,94,95,96,97,98,99,100,101,102,103,104,105,106,107,108,109,110,111,112,113,114,115,116,117,118,119,120,121,122,123,124,125,126,127,128,129,130,131,132,133,134,135,136,137,138,139,140] (Figure 1).

### 3.2. General Characteristics of the Studies

A total of 101 articles were included, of these, 91 were from Brazil, five from Chile, two Colombia, two Mexico and one from Ecuador. The number of participants at the beginning of the studies was 5013 and 4334 at the end (79% women and 21% men), presenting a compliance with the interventions of 86%. Only 35% of the studies had 100% adherence. Three quarters of the studies suffered losses in their sample, the main reasons for withdrawal being categorized as follows: attendance, does not meet the criteria of the study and personal reasons, the latter being the one that was repeated the most (78%). Furthermore, in 15% of the cases the reason for withdrawal was not specified. Regarding the age range, 97% of the studies included participants between 60 and 70 years, 83% between 71 and 80 years, and only 16% included participants older than 80 years. The health condition with the highest prevalence was cardiometabolic diseases. Just over half (58%) of the OAs resided in the community and 44% of the studies detailed information regarding the place of intervention; of these studies, 61% of the interventions occurred at university facilities (Table 2).

### 3.3. Assessment of Risk of Bias and CERT Compliance Level

The risk of bias of the studies was assessed using the Cochrane ‘Cochrane Handbook for Systematic Reviews of experimental study interventions’ tool. The risk of bias was included for each of the items proposed by the manual, excluding the item “other biases”. It was observed that the distribution of biases classified as unclear risk or high risk was similar between the items selection bias, detection bias and attrition bias (40–60% of the studies). However, it was observed that 95% of the studies had unclear risk and high risk of performance bias, and 100% of the studies had low risk of reporting bias (Figure 2).

Nine out of nineteen items compliant (rated “yes” on the CERT items) in at least 75% of the articles (C1 = 90%, C3 = 100%, C4 = 100%, C5 = 93%, C7b = 100%, C10 = 100%, C13 = 100%, C14a = 100%, C16a = 93%). The items categories with the highest level of compliance were those included in the materials (one item, C1 = 90%) and dosage (one item, C13 = 90%) categories. In contrast, the items categories with the lowest level of compliance were those related to the delivery (ten items, C6 = 5%, C7a = 50%, C8 = 21%, C9 = 0% and C11 = 41%) and location (one item, C12 = 40%) categories (Appendix A).

### 3.4. Main Characteristics of the Interventions

95% of the articles had interventions with a duration of 2 to 6 months, 2% less than 2 months and 3% more than 9 months. 95% had a session frequency of 2 or 3 times a week. The sessions had a duration that varied between 30 and 60 min in 93.8% of the articles, taking into account that 20.8% of the studies did not specify this information. In 51% of the studies the sessions were led by physical activity qualified professionals, such as physical education teachers or personal trainers, while 14% of the studies presented interventions led by a health professional, most of whom were a physiotherapist. 35% of the studies did not specify this information. Regarding the training modality, 89% of the studies included interventions based on therapeutic PE, 11% of the studies included interventions based on non-traditional physical disciplines such as Tai Chi, Pilates and dance, 7% included interventions based in exercise with digital support, known as Exergames, 5% included interventions based on exercise complemented with other interventions such as vibration and auriculotherapy and 2% included interventions based on water training, known as hydrogymnastics. Regarding the components of physical fitness addressed during the interventions, 77% of the studies included muscular strength, 47% cardiorespiratory fitness, 27% balance, 14% coordination, 12% flexibility, 7% gait and 5% proprioception. Furthermore, 48% of the studies included a warm-up stage and 34% of the studies included a cool-down stage (Table 3). 

Additionally, the characteristics of multicomponent exercise were analyzed according to the components of physical fitness. Muscle strength was commonly exercised with 1 to 3 sets (61%), 8–15 repetitions (71%), and one-minute rest (18%).The intensity of the exercise was controlled with scales of perception of effort (30%) and multifunctional machines were used to train (65%). In relation to cardiorespiratory fitness, this was developed mainly on a treadmill (39%), for 20 min or more (61%). The intensity of the exercise was controlled through the heart rate (52%). With regard to flexibility, this was exercised through static stretching (25%) and was controlled through the time performed (42%), With regard to gait, coordination, and proprioception, these components of fitness were trained primarily through circuits (56%, 50% and 40, respectively).

### 3.5. Outcome Variables Analyzed

Outcome variables were grouped and described in three broad categories: (a) physical health, (b) mental health and quality of life and (c) cognitive skills.

(a)Physical health: This category was considered in 100% of the studies and was divided into 10 outcome variables. Of these, the most evaluated was muscle strength (74 of the studies). The following most frequent were: nutritional status and diet, functionality, balance, gait and vital signs, blood tests and others (blood pressure, pain, dyspnea, heart rate variability and blood tests). The least evaluated was coordination (7 studies) (Figure 3a).(b)Mental health and quality of life: The mental health and quality of life categories were considered only in 28% of the studies, and was grouped into nine outcome variables related to emotional, psychological and social well-being. The most evaluated was quality of life, included in 17 studies, followed by depression and fear of falling, evaluated in eight and six studies, respectively (Figure 3b).(c)Cognitive Skills: the category of cognitive skills was considered only in 11% of the studies and 12 outcome variables were grouped. The most evaluated was language, included in six of the studies, followed by memory, attention and executive function, evaluated in five studies (Figure 3c).

### 3.6. Assessment Instruments Used

The evaluations were grouped and described in three broad categories: (a) physical health, (b) mental health and quality of life, and (c) cognitive skills.

(a)Physical health instruments: 63 instruments were used, which were grouped into ten categories (strength, flexibility, cardiorespiratory fitness, walk test, balance, chair test, step test, risk of falls, functionality and body composition), being the most used instruments the maximum repetition to measure muscle strength (33 studies), the sit-to-stand test in tests that use a chair (24 studies), the timed up and go in tests of risk of falling (20 studies) and the test of 6-min walk in walking tests (17 studies) (Figure 4a).(b)Mental health and quality of life instruments: 18 instruments were used, which were grouped into four categories (quality of sleep, suspected depression, quality of life and others, which included instruments that evaluated affectivity, efficacy, mood, feelings, entertainment and perception barriers). The most widely used instruments were the Falls Efficacy Scale International (FES-I), the World Health Organization Quality of Life (WHOQoL), the Geriatric Depression Scale (GDS) and the Yesavage Geriatric Depression (YGDS) (Figure 4b).(c)Cognitive skills instruments: 22 instruments were used that were grouped into two categories (cognitive skills and suspected dementia). Of all the instruments, the most used were the Mini Mental State Examination (MMSE) in five studies and the Montreal Cognitive Assessment (MoCA) (Figure 4c).

### 3.7. Effects by Sex and Type of Intervention

Regarding the effects of physical exercise interventions according to sex, in the physical health category, more than 50% of the articles with female-only samples presented significant effects in six of nine outcomes. More than 50% of the articles with male-only samples presented significant effects in four of seven outcomes, and more than 50% of the articles with mixed samples (both sexes) presented significant effects in three of nine outcome variables. In the mental health category, more than 50% of the articles with female-only samples presented significant effects in two of five outcomes. Articles with male-only samples did not include mental health outcome variables. More than 50% of the articles with mixed samples (both sexes) presented significant effects in four of six outcomes. Regarding the cognitive skills category, more than 50% of the articles with female-only samples presented significant effects in four of 10 outcomes. 100% of the articles with male-only samples presented significant effects in eight of eight outcome variables. More than 50% of the articles with mixed samples (both sexes) presented significant effects in five of 10 outcomes (Figure 5).

Regarding the effects according to the type of intervention, in the physical health category, more than 50% of the therapeutic physical exercise-based interventions presented significant effects in seven of nine outcomes. More than 50% of the hydrogymnastics-based interventions had significant effects in three of six outcomes. More than 50% of the non-traditional physical disciplines-based interventions had significant effects in two of nine outcomes. More than 50% of the digitally supported exercise-based interventions had significant effects in one of nine outcomes. More than 50% of the exercise-based interventions complemented with other interventions had significant effects in two of five outcomes. Regarding the mental health category, more than 50% of therapeutic physical exercise-based interventions presented significant effects in six of nine outcomes. There was no hydrogymnastics-based intervention that evaluated mental health outcomes. More than 50% of the non-traditional physical disciplines-based interventions had significant effects in one of three outcomes. More than 50% of the digitally supported exercise-based interventions had significant effects in one of three outcomes and 100% of the exercise-based interventions complemented with other interventions had significant effects on fear of falling. Regarding the category of cognitive skills, more than 50% of the therapeutic physical exercise-based interventions presented significant effects in 11 of 12 outcomes. There was no hydrogymnastics-based intervention that evaluated cognitive skills outcomes. More than 50% of the non-traditional physical disciplines-based interventions had significant effects in one of nine outcomes. 100% of the exercise-based interventions complemented with other interventions had significant effects in two of three outcomes (Figure 5).

## 4. Discussion

The main results of this systematic review focused on five large areas: general characteristics, risk of bias and level of compliance with the CERT of the articles, characteristics of the interventions, outcome measures, and instruments used for their analysis. Its importance and possible implications are discussed below.

### 4.1. General Characteristics of the Articles Reviewed

Regarding the geographic location, the interventions occurred in only five Latin American countries, most of them in Brazil. This could be associated with the fact that Brazil is the country with the largest population in Latin America, and the sixth in the world [141], being also one of the Latin American countries that invests the most in developing and publishing research in the area of biological and medical sciences [142,143]. Regarding the characteristics of OA, female participants predominated (79%), which is striking considering that South American population surveys position women as more physically inactive than men [144]. Adherence was higher than 80% in 75% of the studies, which suggests a high degree of commitment to this type of intervention and contrasts with the evidence that indicates that the practice of AP decreases over the years [145]. In addition, OAs were mainly recruited from the community and most of the interventions were carried out in university facilities, suggesting that OA living in the community are willing to participate in PE-related activities. This could motivate researchers to develop interventions that include more OAs as study subjects in future research. Although the evidence is clear that the benefits of PA and PE are independent of age and health status [11], it is interesting that only 16% of the studies included people over 80 years of age, which could be associated with the higher prevalence of comorbidities at that age [146]. This would generate a higher risk of unwanted side effects related to exercise, although if the training program is prescribed properly, the risk of side effects should be like those that occur in OAs below 80 years [9,10]. Regarding the health condition of the OAs, the comorbidities with the highest prevalence were neurodegenerative and cardiometabolic diseases, which is related to the increase in NCDs worldwide [13].

### 4.2. Evaluation of the Methodological Quality of the Studies and CERT Compliance

From the results found, it stands out that 95% of the studies presented unclear risk and high risk in performance bias. This point is relevant as it is related to the blinding of the study participants and staff, and although the Cochrane manual states that simply blinding does not ensure successful blinding, it also states that the lack of blinding in this item could produce a bias by affecting the results of the participants [39]. In any case, the predominant classification in this item was unclear risk, which is associated with a lack of information on this bias by the authors, rather than a possible bias in the results. On the contrary, 100% of the studies presented a low-risk classification in the reporting bias item. This could be due to the fact that all the studies had an experimental and randomized methodological design (RCT), which corresponds to the best level of evidence in quantitative studies. Only nine out of nineteen items compliant in at least 75% of the RTC analyzed. Furthermore, the items categories with the highest level of compliance were those included in the materials and dosage categories. Conversely, the item categories with the lowest level of compliance were those related to the delivery and location categories. Similar results on the level of compliance had already been observed in another systematic review. This could be because the CERT was developed in 2016 with the goal of making exercise-based clinical trials transparent and replicable and have not yet been sufficiently assimilated by researchers [147].

### 4.3. Main Characteristics of the Interventions

Most of the interventions had a duration of 2 to 6 months, with a frequency of 2 to 3 times a week and a duration of 30 to 60 min per session, and were directed by qualified professionals, characteristics that are related to the WHO recommendations [19,20] and to the GPAR [21,22,23,24,25,26,27,28,29,30,31,32,33,34,35]. 89% of the interventions were based on traditional PE, while 11% of the studies included interventions based on non-traditional disciplines such as Tai Chi and Pilates, disciplines that in recent years have increased their adherence and level of evidence [44,81,111,125,148] being even included by some countries in their GPAR [21,27,29,32,33]. 7% of the studies included exercise-based interventions with digital support, known as Exergames [74,78,117,120,126] a modality not so well known today, which has been shown to be effective in improving balance and mobility function in OA [149,150]. On the other hand, some authors suggest that more research is lacking regarding this type of interventions and the effects they propose [151]. 2% of the studies included interventions based on hydro gymnastics [125,127], and although few studies have covered this type of exercise, it has shown positive effects on the physical functioning of OAs [152]. Finally, 5% included exercise interventions supplemented with other interventions such as vibration and auriculotherapy [48,57,82,140]. Regarding the vibration modality, benefits have been shown in physical performance [153] and in balance and gait of OAs [154], in addition to being an effective intervention to reduce the risk of falls [155]. However, some authors suggest that more evidence is needed to know which is the most effective vibration modality [156]. This indicates that although interventions based on traditional exercise predominate (mainly resistance training [157] and aerobic training [158]), whose effects have been widely supported, non-traditional interventions have been developed during the last decade that have shown to be as effective as traditional exercise interventions [48,57,82,140]. Regarding the components of the interventions, 77% considered the muscle strength component as the main training variable and 47% the aerobic exercise, the latter being the first recommendation by the WHO and the GPAR. This may be due to the fact that muscle strength training, associated with an improvement in muscle quality in OAs, is more closely related to improvements in functionality, decreased risk of falls, and health-related improvements in quality of life [159,160], variables in which aerobic training has a minor effect [160]. At the same time, the small number of studies that consider the training of balance, coordination and proprioception components within their interventions is striking, given that alteration of these components increases the risk of falls. This variable that is of great interest to OAs, because falls are associated with a decrease in functionality, a higher level of dependency and a higher risk of mortality [12,13,14,15,16]. Lastly, the flexibility component was considered only in 7% of the studies, a variable that is not specifically found within the WHO recommendations, and is only found in some of the GPAR. This contrasts with the fact that flexibility exercises have been shown to improve aspects of physical health and mental well-being in OAs [36,161].

A fifth of the studies included in this review developed combined PE interventions, modalities known as concurrent training and multicomponent training, which had positive effects on the variables studied. In particular, the multicomponent training modality, which has important scientific support in terms of its effectiveness [162,163], is being strongly recommended in the European continent for the development of PA in OA [164]. In addition, it could be that in the near future, the recommendations proposed by the WHO will undergo modifications, proposing multicomponent training as a more effective alternative than training each component of physical fitness in isolation in OA.

### 4.4. Outcome Variables Studied in the Articles and Instruments Used in Evaluations

The aging process is not only associated with deterioration in physical skills, but also with disorders of mental health and cognitive skills [165]. Along these lines, although the interventions were based on PE that considered physical health variables in 100% of the studies, 28% of the studies also considered the effects of these interventions on mental health variables and 11% on cognitive skills. Even though the percentage is low, this is related to understanding how exercise is a therapeutic agent with multiple benefits in the health of OA [17].

Sixty-three instruments were used to evaluate the physical health variables, 18 to evaluate the mental health variables, and 22 to evaluate the cognitive skills variables. This highlights the great diversity of instruments used to identify the effects of PE in OA, and although there are some instruments that are used more frequently, this highlights the lack of consensus that exists for the evaluation of the different variables.

### 4.5. Effects by Sex and by Type of Intervention

Regarding the effects of interventions by sex, positive effects were observed in a greater number of physical health outcome measures in male-only samples, in a greater number of mental health outcome measures in mixed samples, and in a greater number of cognitive function outcome measures in single samples of men. Older women are known to have lower levels of PA and fitness than older men [144]. Considering this baseline condition, a PE-based intervention could show greater benefits for women in the different physical health outcomes [166], Furthermore, it has been observed that mixed sample generates greater collaboration, motivation and challenge for OA of both genders [167]. 

In relation to the effects according to different types of exercise, positive effects were observed in a greater number of physical health, mental health and cognitive function outcome measures in therapeutic PE-based interventions. This could be related to the fact that this type of intervention is usually implemented as an intervention for OA with altered physical health status. In addition, this type of intervention is more structured, individualized and with specific objectives for each participant, allowing greater control of exercise dose such as intensity, frequency, length and training progression. Furthermore, therapeutic PE-based interventions commonly included in the same session a greater number of components of physical condition [45,100,103,131], than those interventions based on non-traditional disciplines. Hydro-gymnastics and with digital support, which although have shown positive effects in some interventions, some authors agree that evidence to support their effects is lacking [151,156]. In particular, therapeutic PE-based interventions seem to improve muscle strength in a greater number of RTC. This finding could be associated with the fact that other types of exercise such as hydro-gymnastics-based interventions and digitally supported exercise-based interventions consider very light workloads and are more focused on other fitness qualities, such as aerobic endurance, balance and coordination [151]. 

Additionally, the component of physical fitness with the most positive effects was balance, regardless of sex and type of intervention. The interventions were mainly with a duration of 3 months, a frequency of 3 times a week, with 30–60 min sessions. OAs often have problems with balance [168]. Thus, PE-based interventions could promote improvements in these components of physical fitness [169,170]. 

Particularly, quality of life improved regardless of the type of exercise trained, which reinforces the positive effect of PE on health-related quality of life of the elderly [171,172]. In cognitive skills, therapeutic PE-based interventions, presenting effectiveness greater than 60% of the articles in 11 of the 12 analyzed outcomes. It should be noted that few studies analyzed mental health and cognitive skills outcomes, most of which were therapeutic PE-based interventions. This finding suggests a lack of evidence on mental health and cognitive skills outcomes that should be explored in future research.

### 4.6. Limitations

Although articles from other countries such as Chile, Mexico, Ecuador and Colombia were considered, most of the included studies were developed in Brazil, which does not allow a complete overview in relation to PE interventions for OAs in all of Latin America. This in turn could be considered a challenge for researchers in the rest of the Latin American countries in terms of developing similar research in their respective countries. On the other hand, we restricted the search the last 5 years to find the latest and updated available evidence. We are aware that there may be high-quality evidence in previous years that was not included. Another limitation of this review was the quality of the RTCs included. Thus, a high percentage of RCTs had unclear risk of bias in selection bias, performance bias, detection bias, and attrition bias. In addition, 52.6% of the CERT items showed a low percentage of articles that complied with describing and detailing the CERT categories. Furthermore, this scoping review lacked meta-analysis due to the studies’ heterogeneity, so there is only a qualitative analysis of the phenomenon studied.

### 4.7. How Does this Literature Review Contribute to the Existing One?

This systematic review provides a broad and updated view of the characteristics of the PA and PE interventions that are being developed in Latin America, thus allowing the generation of a profile, outcome variables, and evaluation instruments used. On the other hand, this review allowed to identify that, although there are certain similarities between the recommendations proposed by the WHO and the GPAR of the different Latin American countries, there are also differences, for example, in the type of exercise, its frequency and its duration. Therefore, this review could serve to determine the more and less frequent characteristics of the interventions used by Latin American researchers. 

This review also provides updated information derived from RTC on the characteristics of PE based-interventions, as well as the most frequently used outcome measures and instruments that should serve to help exercise professionals prescribe exercise for older adults.

Additionally, it allowed the understanding of new PE modalities that are being implemented and their characteristics, being able to encourage the development of future studies that are based on non-traditional PE interventions, in little-explored age groups and health conditions.

## 5. Conclusions

A total of 101 articles were included (90% focused on Brazil), with a total sample of 5013 OA (79% women). 97% of the studies included participants between 60–70 years of age. The studies had an average adherence of 86%. The main cause of withdrawal was personal reasons (78% of cases). The studies were mainly carried out with OAs who lived in the community with the diagnosis of some disease, with cardiometabolic diseases being the most prevalent. A large percentage of studies had an unclear risk of bias in the items: selection, performance, detection and attrition. Furthermore, a heterogeneous level of compliance was observed in the CERT items. The interventions were mainly based on therapeutic PE, lasted between 2–6 months, with a frequency of 2–3 times a week, with a duration of sessions between 30–60 min and were led by professionals of PA sciences. For the most part, there was no warm-up stage before exercise and cool-down stage after exercise. As well as the components of physical fitness that were exercised the most were muscle strength and cardiorespiratory fitness. 

This systematic review provides a broad and updated view of the characteristics of PE-based interventions that are being developed for OAs in Latin America. The findings will be useful for prescribing future PE programs for OAs, being able to encourage the development of future studies in this area of knowledge.

## Figures and Tables

**Figure 1 ijerph-18-02812-f001:**
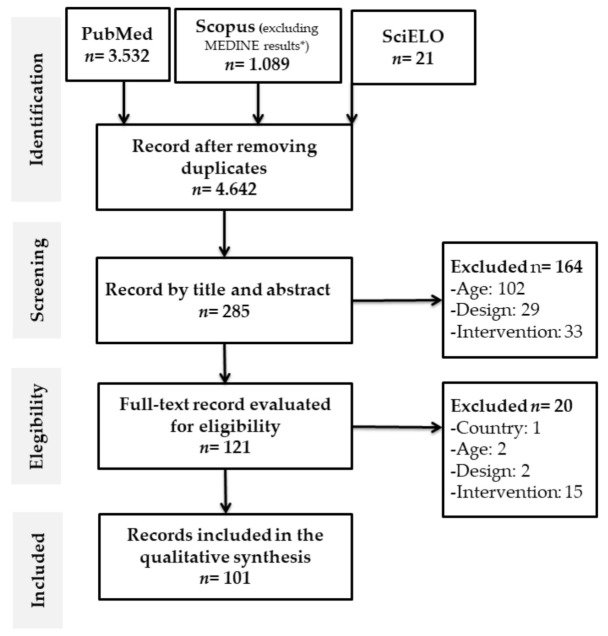
Literature search Flow chart. * Whole search strings for all databases are presented in the Appendix A.

**Figure 2 ijerph-18-02812-f002:**
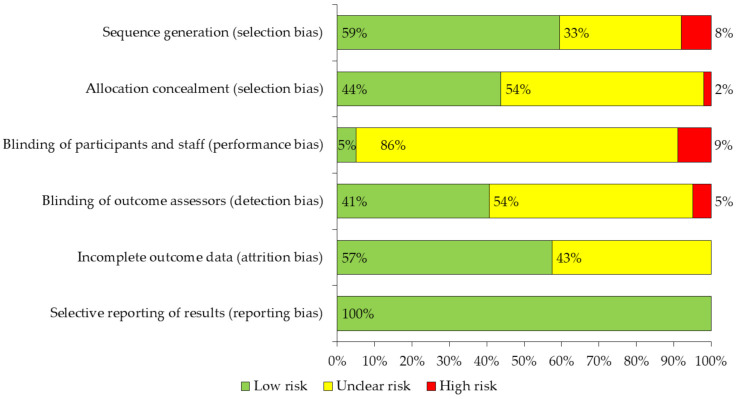
Evaluation of the methodological quality of the reviewed studies.

**Figure 3 ijerph-18-02812-f003:**
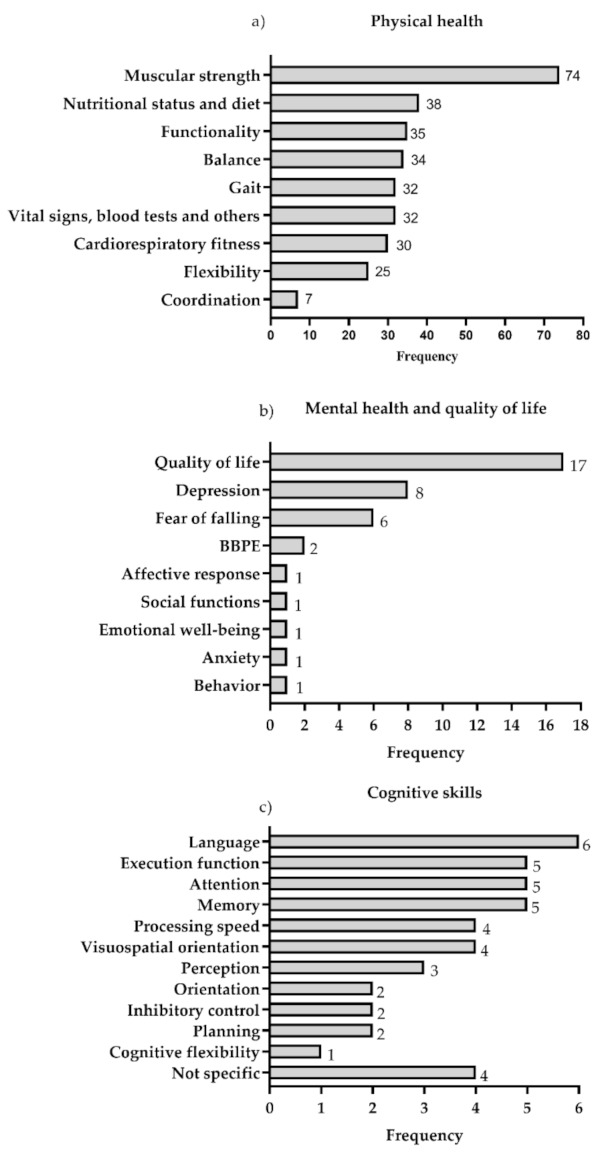
Variables of (**a**) physical health, (**b**) mental health and quality of life and (**c**) cognitive skills. The data are presented in absolute frequency. BBPE: Barriers and Benefits Perceived for the Execution of the Exercise.

**Figure 4 ijerph-18-02812-f004:**
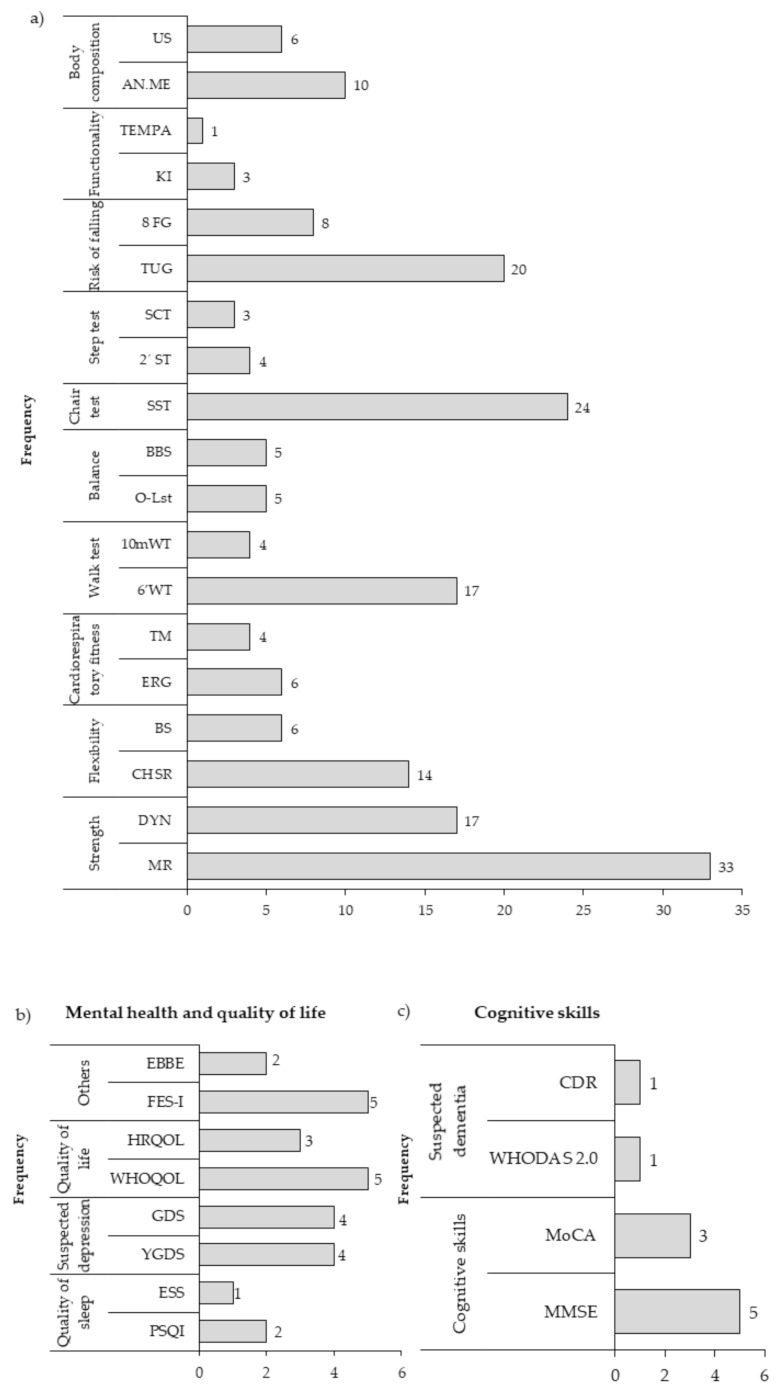
Instruments for measuring (**a**) Physical Health variables. Strength: MR, Maximum repetition; DYN, Dynamometer. Flexibility: CHSR, Chair sit and reach; BS, Back Scratch. cardiorespiratory fitness: ERG. Ergospirometry; TM, Treadmill. Walk test: 6’WT, 6-min walk test; 10 mWT, 10-m walk test. Balance: O-Lss, One-legged stand test; BBS, Berg Balance Scale. Chair test: SST, Sit to Stand Test. Step Test: 2’ST, 2 min Step Test, SCT, Stair Climb Test Risk of falling: TUG, Time up and go; 8FG, 8-fit and go. Functionality: KI, Katz index; TEMPA, Test d’Evaluation des Membres Supérieurs de Personnes Âgées. Body composition: ANT. ME., Anthropometric measures; US, Ultrasound. (**b**) Mental health and quality of life variables. Sleep quality: PSQI, Pittsburgh Sleep Quality Index; ESS, Epworth sleepiness scale. Suspected depression: YGDS, Yesavage Geriatric Depression Scale; GDS, Geriatric Depression Scale. Quality of life: WHOQOL, The World Health Organization Quality of Life; HRQOL, Health Related Quality of Life. Others: FES-I, Falls Efficacy Scale International; EBBE, Exercise Benefits Barriers Scale. (**c**) Cognitive skills variables. Cognitive skills: MMSE, Mini-Mental State Examination; MoCA, Montreal Cognitive Assessment. Suspected dementia: WHODAS 2.0, World Health Organization Disability Assessment Schedule; CDR, Clinical Dementia Rating. The data are presented in absolute frequency.

**Figure 5 ijerph-18-02812-f005:**
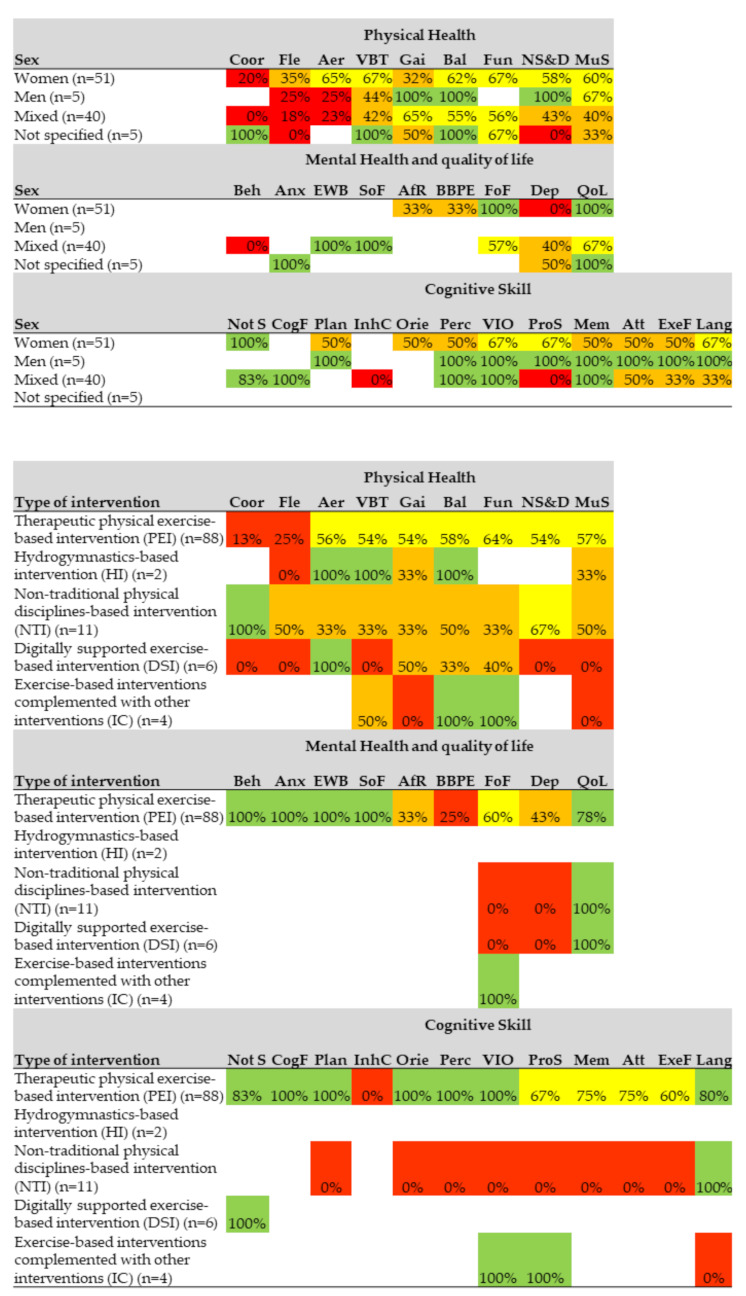
Effects by sex and by type of intervention on the outcomes of physical health and quality of life, mental health and cognitive skills. The percentage of articles that present significant effects on each health outcome measures is shown. 0–25% red, 26–50% orange, 51–75% yellow, 76–100% green and white: no studies. Physical health Coor: Coordination, Fle: Flexibility, Aer: cardiorespiratory fitness, VBT: Vital signs, blood tests and others, Gai: Gait; Bal: Balance, Fun: Functionality, NS&D: Nutritional status and diet, MuS: Muscular strength. Mental health and quality of life: Beh: Behavior, Anx: Anxiety, EWB: Emotional well-being, SoF: Social functions, AfR: Affective response, BBPE: Barriers and Benefits Perceived for the Execution of the Exercise, FoF: Fear of falling, Dep: Depression, QoL: Quality of life. Cognitive skills Not S: Not specific, CogF: Cognitive flexibility, Plan: Planning, InhC: Inhibitory control, Orie: Orientation, Perc: Perception, ViO: Visuospatial orientation, ProS: Processing speed, Mem: Memory, Att: Attention, ExeF: Execution Function, Lang: Language.

**Table 1 ijerph-18-02812-t001:** Inclusion criteria of the studies.

Criterion	Description
- Type of intervention - Single intervention - Duration	- Resistance, strength, multicomponent, concurrent, multidomain, HIIT or neuromotor, or other related to physical activity or exercise. - Only intervention based on physical activity or exercise (no other interventions). - Duration of at least four weeks.
- Age range	- Seniors, 60 years or older.
- Physical health - Mental health - Cognitive skills	- Physical fitness variables: balance, muscular strength, cardiorespiratory fitness, flexibility, proprioception, agility, other. - Psychological variables: depression, happiness, well-being, quality of life, anxiety, other. - Cognitive variables: memory, perception, language, attention, concentration, other.
- Article type	- Original article, with experimental design and random assignment.
- Country of origin of the population	- Latin American countries only.

**Table 2 ijerph-18-02812-t002:** Characteristics of the articles.

Ref.	Authors (Year) Country	Initial/Final Sample	Percentage that Ended the Study (%)	Withdrawal				Age Range			Sex		Health				Recruitment				Place of Intervention	
				Reason				(Years)			♀	♂	Condition									
				AT	PR	NMC	NS	60–70	71–80	80 o+	(%)	(%)	CD	NDD	CMD	CA	COM	HC	INST	HC	SC	UC
[40]	Queiroz J. (2016) Brazil	62/62	100					1	1		0	100	-	-	-	-	1					
[41]	Antunes A. (2015) Brazil	45/45	100					1	1		0	100	-	-	-	-	1					
[42]	Santos S. (2017) Brazil	40/26	65	1	1			1	1		31	69		1				1		1		
[43]	de Oliveira R. (2017) Brazil	24/23	96		1			1	1	1	56	44		1				1				1
[44]	Campos L. (2016) Brazil	32/32	100					1			100		-	-	-	-	1			1		1
[45]	Mazinifilho M. (2017) Brazil	79/79	100					1	1		100										1	
[46]	Teodoro J. (2019) Brazil	36/36	100					1	1			100	-	-	-	-	1					
[47]	Santos G. (2019) Brazil	34/18	53	1				1	1		100			-			1	1				
[48]	Dueñas E. (2019) Colombia	125/105	84	1	1			1	1	1	84	16	1		1		1	1				
[49]	Torres A. (2019) Brazil	64/56	88		1			1	1		91	9					1					
[50]	Arantes P. (2015) Brazil	30/28	93		1			1	1	1	100			-				1				
[51]	Lima L. (2015) Brazil	44/44	100					1	1		84	16		-	1						1	
[52]	de Oliveira D. (2019) Brazil	24/19	79		1			1	1		100											
[53]	da Silveira Ch. (2019) Brazil	60/52	87		1	1		1	1	1	77	23		1			1				1	
[54]	do Nascimento M. (2019) Brazil	62/45	72	1	1			1	1		100	0					1					
[55]	Oliveira F. (2016) Brazil	25/25	100					1	1						1		1					
[56]	Galvao M. (2019) Brazil	30/24	80	1	1			1	1		100				1							1
[57]	Carvalo R. (2018) Brazil	22/21	95		1			1	1		75	25		-				1		1		
[58]	Tiggemann C. (2016) Brazil	30/25	83	1	1		1	1	1		100										1	
[59]	da Silva R. (2017) Brazil	30/30	100					1	1		100						1					1
[60]	Taglietti M. (2018) Brazil	60/49	81	1	1			1	1		68	32	1					1		1		
[61]	Franco M. (2016) Brazil	82/71	87	1	1			1	1		93	7		-			1					1
[62]	Batisti C. (2018) Brazil	45/37	82		1			1	1	1	NS	NS							1	1		
[63]	Lopez N. (2015) Chile	80/60	75	1	1			1	1	1	68	32		1				1			1	
[64]	Shiguemitsu F. (2018) Brazil	37/31	84	1							100											1
[65]	Ortiz-Ortiz M. (2019) Mexico	50/50	100					1	1	1	60	40							1	1		
[66]	Mansur J. (2016) Brazil	35/35	100					1	1		100	0			-		1					
[67]	Clemente A. (2018) Brazil	41/35	86		1	1		1	1		66	34			1		1					1
[68]	Rodrigues-Krause J. (2018) Brazil	30/27	87		1			1			100		1		-		1				1	
[69]	Ferrari R. (2016) Brazil	24/23	96		1			1	1			100	-	-	-	-						
[70]	Ramirez-Campillo R. (2016) Chile	24/24	100					1	1	1	100	0					1					1
[71]	de Resende A. (2018) Brazil	32/32	100					1	1		100		-	-	-		1					
[72]	Cavalcante E. (2018) Brazil	63/57	91		1			1	1		100				1		1					
[73]	Pieta C. (2015) Brazil	26/19	73		1			1	1		100	0										
[74]	Henrique P. (2019) Brazil	31/31	100					1	1	1	55	45			1			1		1		
[75]	Ramírez-Villada J. (2019) Colombia	60/47	79	1				1			100		1									1
[76]	Gomes A. (2019) Brazil	47/47	100					1	1		100			-								
[77]	Feitosa N. (2016) Brazil	30/23	77	1				1			100						1					
[78]	Ribeiro J. (2018) Brazil	50/46	92		1			1	1		74	26					1			1		
[79]	dos Santos L. (2018) Brazil	39/39	100					1	1		100				-		1					
[80]	Gomeñuka N. (2019) Brazil	33/26	79		1		1	1	1		73	27			1		1					1
[81]	Campos L. (2015) Brazil	32/32	100					1			NS	NS	-				1					1
[82]	de Souza R. (2018) Brazil	42/27	64	1	1			1	1		100						1					
[83]	Macedo L. (2018) Brazil	23/19	83	1	1	1		1			100						1					1
[84]	Botton C. (2018) Brazil	44/26	59	1	1			1	1		41	59			1		1			1		1
[85]	Bonadias. A. (2016) Brazil	133/133	100					1	1		100		1		1	1	1					1
[86]	Barbosa A. (2015) Brazil	30/30	100					1	1		100						1					1
[87]	Rodrigues W. (2015) Brazil	47/40	85		1			1	1		70	30	-	-	-	-	1	1				1
[88]	Gallo L. (2015) Brazil	31/26	84	1				1	1		100						1					1
[89]	Ruaro M. (2019) Brazil	40/33	83		1	1		1	1		100						1					
[90]	Da silva C. (2018) Brazil	58/51	88		1			1	1		59	41	1				1			1		
[91]	Mirando A. (2020) Chile	21/12	57				1	1			86	14	-				1					
[92]	Cadore E. (2018) Brazil	65/52	80		1			1	1			100	-	-	-	-	1					
[93]	de Resende A. (2016) Brazil	55/44	80				1	1	1		100						1					
[94]	Silva I. (2018) Brazil	48/43	90				1	1	1		39	61	1									
[95]	Rabelo M. (2019) Brazil	39/39	100					1	1		74	26			1			1				
[96]	Ramirez-Campillo R. (2018) Chile	74/52	70	1				1	1		100		-	-	-							
[97]	Brandao G. (2018) Brazil	131/125	95		1			1	1		88	12					1				1	
[98]	Lopez J. (2017) Mexico	31/26	84		1			1	1		100		-	-			1					1
[99]	Medeiros L. (2018) Brazil	78/71	91	1				1	1		77	23		-			1					
[100]	Vargas M. (2019) Ecuador	50/50	100					1			30	70			1					1		
[101]	Covolo-Scarabottolo C. (2017) Brazil	35/30	86				1	1	1	1	53	47							1	1		
[102]	Damorim I. (2017) Brazil	64/55	86	1	1			1			71	29			1		1					
[103]	Leal L. (2019) Brazil	54/54	100					1			50	50		1					1			
[104]	Souza D. (2019) Brazil	25/21	84		1			1	1		100		-	-	-	-	1					
[105]	Santos G. (2015) Brazil	70/62	86				1	1	1	1	60	40		1					1			1
[106]	Moreira N. (2018) Brazil	46/45	98		1				1	1	100		-	-	-	-			1			
[107]	Martinez A. (2018) Chile	33/33	100					1	1	1	39	61							1			
[108]	Santana M. (2016) Brazil	23/16	70	1	1			1	1		87	13										1
[109]	Gomeñuka N. (2020) Brazil	33/26	79		1			1	1		72	28					1					1
[110]	Coelho-Júnior H. (2019) Brazil	45/36	80		1			1	1		100		-	-	-		1					
[111]	Silva M. (2016) Brazil	78/45	58		1	1		1	1		82	18										
[112]	Gambassi B. (2015) Brazil	17/16	94	1				1			100						1					
[113]	Tomeleri C. (2016) Brazil	38/35	92	1				1	1		100				1		1					
[114]	Alex S. (2015) Brazil	30/30	100					1	1		100				-		1					1
[115]	Cunha P. (2019) Brazil	48/48	100					1	1		100				-							
[116]	Ribeiro. S. (2017) Brazil	76/68	89		1			1	1		100						1					1
[117]	Alcantar T. (2019) Brazil	33/33	100					1	1		NS	NS		1			1					
[118]	Tomeleri M. (2018) Brazil	53/45	85		1			1	1		100				-							
[119]	Oliveira-Dantas F. (2020) Brazil	25/25	100					1	1		100				1		1					
[120]	Lopez P. (2016) Brazil	55/37	67				1	1	1		100			-	-		1					
[121]	da Silva P. (2015) Brazil	20/20	100					1	1		65	35			1			1				
[122]	Alves W. (2019) Brazil	32/28	88		1			1	1		50	50		1				1				
[123]	Morales F. (2018) Brazil	35/35	100					1			NS	NS		1			1					1
[124]	Rosa C. (2017) Brazil	92/55	60	1	1		1	1	1		100						1					
[125]	Rodacki A. (2017) Brazil	38/30	79				1	1	1		100						1					
[126]	Aragao-Santos J. (2019) Brazil	44/44	100					1	1		100											
[127]	Dominguez D. (2018) Brazil	72/62	86	1	1			1	1		40	60		1				1		1		
[128]	Sbardelotto M. (2017) Brazil	55/55	100					1	1			100					1					
[129]	Moreira H. (2015) Brazil	51/51	100					1			100											
[130]	De Carvalho I. (2018) Brazil	20/20	100					1	1		100							1				
[131]	Mendes M. (2017) Brazil	420/376	90	1				1	1		100		-									
[132]	de Oliveira F. (2019) Brazil	56/46	82	1	1			1	1	1	59	41		1			1					1
[133]	Lixandrao M. (2016) Brazil	14/14	100					1	1		43	57	-		-							
[134]	Ribeiro A. (2016) Brazil	29/25	86		1			1	1		100						1					1
[135]	Silveira Y. (2019) Brazil	83/40	48	1	1			1	1		100							1				
[136]	Monteiro-junior R. (2017) Brazil	29/11	38		1					1	33	67					1		1			
[137]	Chaves M. (2017) Brazil	36/36	100					1	1		100		1				1					1
[138]	Ribeiro S. (2018) Brazil	48/33	69		1			1			100											
[139]	de Oliveira V. (2019) Brazil	52/43	83		1			1			NS	NS					1					
[140]	Simão A. (2019) Brazil	15/15	100					1	1	1	100		1					1				

1: registered data, Withdrawal reasons: AT, Attendance; PR, Personal reasons; NMC, does not meet criteria; NS, Not specified, Blank space, Does not meet criterion. Age range: Blank space, does not meet criterion. Sex: NS, Not specified. Health condition: CD, Chronic disease; NDD, Neurodegenerative disease; CMD, cardio metabolic disease; CA, Cancer; -, Exclusion criterion of the study. Recruitment: COM, Community; HC, Health center; INST, Institutionalized. Place of intervention: HC, Health center; SC, Sports center; UC, University center.

**Table 3 ijerph-18-02812-t003:** Characteristics of the interventions.

Ref.	Authors	Intervention	Intervention	Session	Professional			Type of Intervention					Components and Time of Each Intervention								
		Duration	Frequency	Duration	in Charge			(Experimental Group)					WS	CS	AER	STR	FLE	GAI	COO	BAL	PRO
		(Month)	(Sessions/Week)	(min)	HP	QP	NS	PEI	HI	NTI	DSI	IC			min	min	min	min	min	min	min
[40]	Queiroz J.	3	2	90			1			EG			1	1							
[41]	Antunes A.	6	3	20–60			1	EG							20–60	NS					
[42]	Santos S.	2	2	60	1			EG1										NS	NS	NS	NS
								EG2								NS	NS				
[43]	de Oliveira R.	6	2	60			1	EG1					1	1		NS		NS		NS	NS
								EG2					1	1		50					
[44]	Campos L.	3	2	60	1					EG1											
								EG2						1			60				
[45]	Mazinifilho M.	3	3	50/50		1		EG1					1	1	15	15					
								EG2					1	1	20	20					
[46]	Teodoro J.	5	2	65–85		1		EG1					1		20–35	40					
								EG2					1		20–35	40					
								EG3					1		20–35	40					
[47]	Santos G.	3	3	40		1		EG1					1	1	NS	20				NS	
								EG2					1	1	10*	20				10*	
								EG3					1	1	10*	20				10*	
[48]	Dueñas E.	2	1	60		1		EG1								NS			NS	NS	
										EG2											
												EG3									
[49]	Torres A.	6	3				1	EG1								NS					
								EG2								NS					
[50]	Arantes P.	3	2	60	1			EG												60	
[51]	Lima L.	2.5	3	NS			1	EG1					1	1	30						
								EG2					1	1	30	NS					
[52]	de Oliveira D.	3.5		40		1		EG					1	1	NS	NS				NS	
[53]	da Silveira Ch.	6	2	60	1			EG						1	20–30	NS					
[54]	do Nascimento M.	3	2,/3			1		EG1								NS					
								EG2								NS					
[55]	Oliveira F.	2.5	2,−3			1		EG								NS					
[56]	Galvao M.	2	3	60			1	EG1							30	30					
								EG2							30	30					
								EG3							30	30					
[57]	Carvalo R.	2	2	100		1						EG			NS	NS	NS			NS	
[58]	Tiggemann C.	3	2			1		EG1								NS					
								EG2								NS					
[59]	da Silva R.	6	2,/3			1		EG1								NS					
								EG2								NS					
[60]	Taglietti M.	2	2	60	1			EG					1	1	20	15					10
[61]	Franco M.	3	2	60		1				EG											
[62]	Batisti C.	3	3	40		1		EG							10,−20	15–30	10		10	10	
[63]	Lopez N.	6	5	60		1		EG					1		40					10	
[64]	Shiguemitsu F.	14	2	75			1	EG					1		NS	NS	NS		NS	NS	
[65]	Ortiz-Ortiz M.	3	5	40–50		1		EG					1	1	NS		NS			NS	NS
[66]	Mansur J.	2	2	30			1	EG1							15–30	30					
								EG2								45					
								EG3							45						
[67]	Clemente A.	6	2	90		1		EG					1		35	50					
[68]	Rodrigues-Krause J.	2	3	60		1				EG1											
								EG2					1	1	40						
[69]	Ferrari R.	2.5	2,/3			1		EG1							30	NS					
								EG2							30	NS					
[70]	Ramirez-Campillo R.	3	2,/3	60		1		EG1					1			NS					
								EG2					1			NS					
[71]	de Resende A.	2	3	60		1		EG1					1			30			NS	NS	
								EG2					1		15	30			15		
[72]	Cavalcante E.	3	2,/3	30		1		EG1						1		NS					
								EG2						1		NS					
[73]	Pieta C.	3	2			1		EG1					1			NS					
								EG2					1			NS					
[74]	Henrique P.	3	2	30	1						EG										
[75]	Ramírez-Villada J.	8	3	60		1		EG								NS			NE	NE	
[76]	Gomes A.	2–3	3	45		1		EG1					1		15	20			15	15	
								EG2					1		15	20					
[77]	Feitosa N.	3	3	50		1		EG					1		NS	25			NS	NS	
[78]	Ribeiro J.	1.75	2	60	1						EG1										
								EG2					1	1	10	10	10		10	10	
[79]	dos Santos L.	2	3	NS		1		EG1								NS					
								EG2								NS					
[80]	Gomeñuka N.	3	3	30–60		1				EG1											
								EG2					1	1				30–50			
[81]	Campos L.	3	2	60		1				EG											
[82]	de Souza R.	4	3	60		1						EG	1	1		40					
[83]	Macedo L.	2	3	50	1	1		EG								NS					
[84]	Botton C.	3	3	NS			1	EG					1			NS					
[85]	Bonadias A.	6	3	NS		1		EG								NS					
[86]	Barbosa A.	2	3	30			1	EG										30			
[87]	Rodrigues W.	2	2	NS	1	1		EG					1			NS					
[88]	Gallo L.	2	3	40		1		EG									40				
[89]	Ruaro M. F.	3.5	2	NS			1	EG								NS					
[90]	Da silva C. M.	2	3	30–60	1			EG					1	1	25	5–15					
[91]	Mirando A. D.	1.5	2	60	1			EG					1	1	NS					NS	
[92]	Cadore E.	3	2	NS		1		EG1							NS	NS					
								EG2							NS	NS					
								EG3							NS	NS					
[93]	de resende A.	3	3	60	1			EG1								25			15		
								EG2							15	25					
[94]	Silva I.	3	3	60			1	EG1							NS	NS					
								EG2							NS	NS					
								EG3							NS	NS					
[95]	Rabelo M.	3	3	50		1		EG1					1		25	20					
								EG2													
[96]	Ramirez-Campillo R.	3	3	60		1		EG1								50					
								EG2								50					
[97]	Brandao G.	3	3	40	1			EG					1	1	NS	NS	NS		NS	NS	
[98]	Lopez J.	3	5	50			1	EG					1	1	30			NS			
[99]	Medeiros L.	3	3	50		1		EG					1	1	10	NS	10			NS	
[100]	Vargas M.	6	3	30–60		1		EG					1	1	15–40						
[101]	Covolo-Scarabottolo C.	3	2	40–50			1	EG							NS	NS		NS		NS	
[102]	Damorim I.	4	3	30			1	EG1								NS					
								EG2							30						
[103]	Leal L.	6	2	30–40		1		EG								30–40					
[104]	Souza D.	3.5	2	NS		1		EG1								NS					
								EG2								NS					
[105]	Santos G. D.	3	2	NS			1	EG													
[106]	Moreira N.	4	3	50			1	EG					1	1			NS		NS	NS	
[107]	Martinez A.	3	3	63			1	EG					1	1						NS	
[108]	Santana M.	2	3	30		1					EG1										
								EG2							30						
[109]	Gomeñuka N.	2	3	NS			1			EG1											
								EG2							NS						
[110]	Coelho-Júnior H.	4.5	2	40		1		EG1					1			NS					
								EG2					1			NS					
[111]	Silva M.	5	2	60			1			EG						NS	NS				
[112]	Gambassi B.	3	2	NS			1	EG								NS					
[113]	Tomeleri C.	2	3	45		1		EG					1			45					
[114]	Alex S.	4.2	6	NS		1		EG					1	1		NS					
[115]	Cunha P.	3	3	20		1		EG								NS					
[116]	Ribeiro. S.	2	3	NS			1	EG1					1	1		NS					
								EG2					1	1		NS					
[117]	Alcantar T.	5	2	40			1	EG					1	1		NS					
[118]	Tomeleri M.	3	3	NS		1		EG					1	1		NS					
[119]	Oliveira-Dantas F.	2.5	2/3	NS		1		EG								NS					
[120]	Lopez P.	3	3	60		1		EG1								NS					
								EG2								NS					
[121]	da Silva P.	1.5	2	30		1		EG1								NS					
								G2								NS					
[122]	Alves W.	4	2	30–40				EG					1			30–35					
[123]	Morales F.	6	2	30–40			1	EG								NS					
[124]	Rosa C.	6	2	60			1	G1					1	1	40–45	40–45					
								EG2					1	1	40–45	40–45					
[125]	Rodacki A.	2	3	60			1			EG			1	1	NS	NS				NS	
[126]	Aragao-Santos J.	3	3	50		1		EG1					1	1	15	25			15	15	
								EG2					1	1	15	25					
[127]	Dominguez D.	2	3	50		1					EG1			1							
								EG2						1	NS						
								EG3						1				NS	NS	NS	NS
[128]	Sbardelotto M.	2	3	60			1	EG1					1	1	30	30					
								EG2					1	1	15	30					
								EG3					1	1	35						
[129]	Moreira H.	6	3	60			1			EG1											
								EG2							60						
[130]	De Carvalho I.	3	2	30			1				EG1									30	
											EG2									30	
[131]	Mendes M.	3	2	NE			1	EG1								NS					
								EG2								NS					
[132]	de Oliveira F.	3	2	60		1		EG					1	1	20	20					
[133]	Lixandrao M.	2.5	2	NS			1	G					1			NS					
[134]	Ribeiro A.	9	3	NS		1		G1					1			NS					
								G2					1			NS					
[135]	Silveira Y.	4	3	50			1		EG				1	1	15	25					
[136]	Monteiro-junior R.	2	2	30–45			1				EG				NS	NS					
[137]	Chaves M.	3	2	45		1			EG1											NS	
								EG2												NS	
[138]	Ribeiro S.	3	2	NS			1	EG1							NS	NS					
								EG2							NS						
[139]	de Oliveira V.	4	2	NS			1	EG1								NS					
								EG2								NS					
[140]	Simão A.	3	3	NS	1							EG1	1		NS	NS					
												EG2	1		NS	NS					

1: registered data, Professional in charge: HP, Health professional; QP, Physical activity qualified professional. Type of intervention: PEI, Therapeutic physical exercise-based intervention; HI, Hydrogymnastics-based intervention; NTI, Non-traditional physical disciplines-based intervention; DSI, Digitally supported exercise-based intervention; ICI, Exercise-based interventions complemented with other interventions. Components and time of each intervention: WS: warm-up stage; CS: cool down stage; AER, Aerobic; STR, Strength; FL, Flexibility; GAI: Gait; COO, Coordination; BAL, Balance; PRO, Proprioception.

## Data Availability

The systematic review protocol was registered in the PROSPERO repository with the code: CRD42020208833. This information was already provided in the methodology section of the systematic review.

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
