# Peer review of "Characteristics of Physical Exercise Programs for Older Adults in Latin America: A Systematic Review of Randomized Controlled Trials"

_ijerph, 2021, doi:10.3390/ijerph18062812_

Round 1

Reviewer 1 Report

The paper is interesting and well written. In my opinion, there are a few issues that should be revised. The authors may take advantage of some of these comments.

  • Sería recomendable revisar las nuevas recomendaciones de la OMS recientemente publicadas sobre actividad física y actualizarlas en el presente trabajo.
  • It would be advisable to review the new WHO recommendations recently published on physical activity and update them in this study.

  • Why has the search been adapted based on the database?

  • Page 16, Line 265-266. The number of studies in which the sample consists only of women is considerable and relevant, so this statement, "Regarding the characteristics of OA, female participants predominated (79%), and adherence was higher than 80% in 75% of the studies which contrasts with the latest health surveys that place women as more physically inactive than men ", could be misleading.

  • In the discussion, the information related to different areas is detailed, it could be interesting, due to the relevant number of studies only of women, to discuss whether there are differences when the exercise proposals were made only for groups of women.

  • Reference nº10, Chile must be capitalized.

Author Response

It would be advisable to review the new WHO recommendations recently published on physical activity and update them in this study.

Answer: We appreciate the comment. We have reviewed and updated the information regarding the new WHO physical activity recommendations. The text has been written in the Introduction pag 2, lines 63 to 65. Now it reads: Current PA and PE recommendations for aging suggest accumulating a minimum of 150 minutes of moderate aerobic PA or 75 minutes of vigorous aerobic PA and varied multicomponent physical activities aimed to improve functional capacity and prevent falls three or more days a week.

Why has the search been adapted based on the database?

Answer: Thank you very much for the comment. The general search syntax is always the same. It should be considered that the search syntax must be adapted in each database, because each one of them has different additional filters.

Page 16, Line 265-266. The number of studies in which the sample consists only of women is considerable and relevant, so this statement, "Regarding the characteristics of OA, female participants predominated (79%), and adherence was higher than 80% in 75% of the studies which contrasts with the latest health surveys that place women as more physically inactive than men ", could be misleading.

Answer: Thank you very much for the comment. the phrase was unclear. The sentence was redone to clarify the concepts of gender and commitment to physical exercise. Page 18, line 312 to 316.

In the discussion, the information related to different areas is detailed, it could be interesting, due to the relevant number of studies only of women, to discuss whether there are differences when the exercise proposals were made only for groups of women.

Answer: Thanks for the comment, a figure has been included (figure 5).  The effects of the different exercise-based interventions on physical, mental and cognitive skills outcomes are presented by gender. These effects are presented in female-samples only, male-samples only, mixed-samples and without specifying sex samples. A description of the figure was added to the results, then analyzed in the discussion, and finally a synthesis of the findings was added in the conclusion and abstract.

Reference nº10, Chile must be capitalized.

Answer: This error was corrected on page 22, line 501.

Reviewer 2 Report

General Comment:

The authors systematically reviewed the studies of physical exercise (PE)-based interventions for older adults (OA) reported from Latin America and Caribbean countries, with focusing on the main characteristics of the interventions, participants, types of exercise, variables and instruments used, and their methodological quality. As the prescription of physical activity (PA) and PE in OA has heterogeneity, it could have important meanings to synthesize the reported studies. However, authors focused on the variables related to only methodology of the studies, but not on the efficacy of interventions. As a result, author described only repetition of the results in discussion section. It is difficult why authors did not include the efficacy of intervention as a variable. The efficacy of intervention should be included as a variable and should be discussed in the discussion section.

Specific comments:

#1. Page 2, Line 93, Page 3, Line 102, Page 3 Table 1

For PubMed, older adult was searched using the word “aged and 80 and over”, however, the inclusion criteria b) defined as “people over 60 years of age”. Some reports investigate the patients aged between 60 and 80 were not detected by PubMed.

#2. Table 1

Rows of 2Criterion” and “Description” are not aligned correctly.

#3. Table 1 Description

The order of “Psychological variables:---” and “Physical condition variables:---” was reversed.

#4. Figure 1

“Record after removing duplicates” was n=285, “Full-text record evaluated for eligibility” was n=121.

In that case, Number of exclusions should be 164 but not 162. There is an error in the aggregation assumption.

Author Response

The authors systematically reviewed the studies of physical exercise (PE)-based interventions for older adults (OA) reported from Latin America and Caribbean countries, with focusing on the main characteristics of the interventions, participants, types of exercise, variables and instruments used, and their methodological quality. As the prescription of physical activity (PA) and PE in OA has heterogeneity, it could have important meanings to synthesize the reported studies. However, authors focused on the variables related to only methodology of the studies, but not on the efficacy of interventions. As a result, author described only repetition of the results in discussion section. It is difficult why authors did not include the efficacy of intervention as a variable. The efficacy of intervention should be included as a variable and should be discussed in the discussion section.

Answer: Thanks for the comment, a figure has been included (figure 5), The effects of the different exercise-based interventions on physical, mental and cognitive skills outcomes are presented by type of intervention. The different types of intervention are: Therapeutic physical exercise-based intervention (PEI), Hydrogymnastics-based intervention (HI), Non-traditional physical disciplines-based intervention (NTI), Digitally supported exercise-based intervention (DSI), Exercise-based interventions complemented with other interventions (IC). A description of the figure was added to the results, then analyzed in the discussion, and finally a synthesis of the findings was added in the conclusion and abstract.

Specific comments:

#1. Page 2, Line 93, Page 3, Line 102, Page 3 Table 1

For PubMed, older adult was searched using the word “aged and 80 and over”, however, the inclusion criteria b) defined as “people over 60 years of age”. Some reports investigate the patients aged between 60 and 80 were not detected by PubMed.

Answer: The Pubmed database has search filters with different options of age ranges that allow adjusting the search according to the age of the subjects. For this review, the additional filter “age” was used. including, aged: 65+ years and 80 and over: 80+ years. Page 3, line 98.

#2. Table 1

Rows of 2Criterion” and “Description” are not aligned correctly.

Answer: This error was corrected on table 1; page 3, line 111

  1. #3. Table 1 Description

The order of “Psychological variables:---” and “Physical condition variables:---” was reversed.

Answer: This error was corrected on table 1; page 3, line 111

#4. Figure 1

“Record after removing duplicates” was n=285, “Full-text record evaluated for eligibility” was n=121. In that case, Number of exclusions should be 164 but not 162. There is an error in the aggregation assumption.

Answer: Thank you very much for the comment. It was a typing error. This error was corrected. Figure 1 was replaced. Now numbers of records exclusions were 164 but not 162.

Round 2

Reviewer 2 Report

The pointed-out parts have been corrected appropriately.

Author Response

Thank you very much for all your comments. We work hard to correct them.